# Targeted Therapy of HPV Positive and Negative Tonsillar Squamous Cell Carcinoma Cell Lines Reveals Synergy between CDK4/6, PI3K and Sometimes FGFR Inhibitors, but Rarely between PARP and WEE1 Inhibitors

**DOI:** 10.3390/v14071372

**Published:** 2022-06-23

**Authors:** Ourania N. Kostopoulou, Mark Zupancic, Mariona Pont, Emma Papin, Monika Lukoseviciute, Borja Agirre Mikelarena, Stefan Holzhauser, Tina Dalianis

**Affiliations:** 1Department of Oncology-Pathology, Karolinska Institute, Karolinska University Hospital, 171 64 Stockholm, Sweden; Ourania.Kostopoulou@ki.se (O.N.K.); Mark.Zupancic@ki.se (M.Z.); marionapont1998@gmail.com (M.P.); emma.papin@etu.umontpellier.fr (E.P.); monika.lukoseviciute@stud.ki.se (M.L.); borjaam2000@gmail.com (B.A.M.); 2Department of Head-, Neck-, Lung- and Skin Cancer, Theme Cancer, Karolinska University Hospital, 171 64 Stockholm, Sweden

**Keywords:** head neck cancer, HPV, tonsillar cancer, base of tongue cancer, oropharyngeal cancer, FGFR, PI3K, CDK4/6, WEE1, PARP, targeted therapy

## Abstract

Human papillomavirus positive (HPV^+^) tonsillar and base of tongue squamous cell carcinoma (TSCC/BOTSCC) have a favorable outcome, but upon relapse, survival is poor and new therapeutical options are needed. Recently, we found synergistic effects by combining the food and drug administration approved (FDA) phosphoinositide 3-kinase (PI3K) and fibroblast-growth-factor-receptor (FGFR) inhibitors BYL719 and JNJ-42756493 on TSCC cell lines. Here this approach was extended and Cyclin-Dependent-Kinase-4/6 (CDK4/6) and Poly-ADP-ribose-polymerase (PARP) and WEE1 inhibitors PD-0332991, and MK-1775 respectively were also examined. HPV^+^ CU-OP-2, -3, -20, and HPV^−^ CU-OP-17 TSCC cell lines were treated with either BYL719 and JNJ-42756493, PD-0332991 BMN-673 and MK-1775 alone or in different combinations. Viability, proliferation, and cytotoxicity were followed by WST-1 assays and the IncuCyte S3 Live^®^ Cell Analysis System. All inhibitors presented dose-dependent inhibitory effects on tested TSCC lines. Synergy was frequently obtained when combining CDK4/6 with PI3K inhibitors, but only sometimes or rarely when combining CDK4/6 with FGFR inhibitors or PARP with WEE1 inhibitors. To conclude, using CDK4/6 with PI3K or FGFR inhibitors, especially PD-0332991 with BYL719 presented synergy and enhanced the decrease of viability considerably, while although dose dependent responses were obtained with PARP and WEE1 inhibitors (BMN-673 and MK-1775 resp.), synergy was rarely disclosed.

## 1. Introduction

Human papillomavirus positive (HPV^+^) tonsillar and base of tongue squamous cell carcinoma (TSCC/BOTSCC), the dominant oropharyngeal squamous cell carcinoma (OPSCC) subsites, generally have better outcome than corresponding HPV-negative (HPV^−^) cancers [1,2,3,4,5]. Moreover, their incidences have increased in many Western countries [6,7,8,9,10,11]. Today most TSCC/BOTSCC patients receive chemoradiotherapy, irrespective of the HPV status of their tumors [12,13]. This aggressive treatment comes with severe side effects and may not be necessary for all patients with HPV^+^ TSCC/BOTSCC, moreover importantly it has not improved survival for those with worse prognosis either [12,13,14,15].

To develop personalized medicine for these patients, in order to de-escalate or target treatment, attempts have been made to find prognostic biomarkers, of which many earlier on were defined by immunohistochemistry, and more recently by molecular methodology [16,17,18,19,20,21,22,23,24,25,26,27,28]. Recently, phosphatidyl-inositol-4,5-bisphosphate 3-kinase, catalytic subunit alpha (*PIK3CA*) and fibroblast growth factor receptor 3 (*FGFR3*) mutations have frequently been revealed in HPV^+^ TSCC/BOTSCC [26,27]. Furthermore, FGFR3 overexpression was reported in HPV^+^ TSCC/BOTSCC/OPSCC and *PIK3CA* and *FGFR3* mutations or changes in their expression have been associated to poorer prognosis, so these genes could be useful for targeted therapy [27,29,30,31].

In breast cancer with *PIK3CA* mutations and urothelial cancer with *FGFR3* translocations or mutations, Food and Drug Administration (FDA) approved phosphoinositide 3-kinases (PI3K) and fibroblast growth factor receptor (FGFR) inhibitors resp. are used clinically [32,33]. For this reason, we recently investigated the effects of FDA-approved PI3K and FGFR inhibitors (alpelisib (BYL719) and erdafitinib (JNJ-42756493) resp.) in TSCC/BOTSCC cell lines with/without *PIK3CA* and *FGFR3* mutations and found both dose-dependent and synergistic effects [34]. In addition, by administering these inhibitors together with cisplatin and docetaxel, used clinically for TSCC/BOTSCC therapy, various combinatory effects were disclosed [34]. Due to the synergistic effects of PI3K and FGFR inhibitors on the viability and proliferation of HPV^+^ TSCC/BOTSCC lines, we were eager to explore whether these inhibitors could also be joined with others and reveal additional synergistic effects. PI3K and Cyclin-Dependent-Kinase-4/6 (CDK4/6) inhibitor combinations have namely also been shown to be of benefit in some types of breast cancer [35].

Additionally, other reports have shown that Poly-ADP-ribose-polymerase (PARP) and WEE1 inhibitors can have synergistic effects in other tumor types such as, e.g., in triple negative breast cancer with Cyclin E or BRCA1 alterations or in biliary tract cancer [36,37]. Moreover, we too have previously shown favorable effects of PARP inhibitors on some, but not all HPV^+^ TSCC/BOTSCC cell lines [38].

Here, therefore in a number of TSCC cell lines, the effects of the CDK4/6 inhibitor (PD-0332991) combined either with the PI3K or the FGFR inhibitors (BYL719 orJNJ-42 756493 resp) as well as the effects of PARP and WEE1 inhibitors, BMN-673 and MK-1775 resp. alone or combined were explored.

## 2. Materials and Methods

### 2.1. Cell Lines and Seeding

HPV^+^ CU-OP-2 (with both a *PIK3CA* and *FGFR3* mutation), CU-OP-3, CU-OP-20 (with a *PIK3CA* mutation) and HPV^−^ CU-OP-17 provided by N. Powell, Cardiff University UK were grown in Glasgow Minimum Essential Medium (GMEM) (Merk Life Science UK, Limited, Dorset, UK) on 60 Gy irradiated 3T3 fibroblasts as feeder layers as described before [34,38]. For analysis, CU-OP lines (without feeders), with 7500 cells/well, were seeded in 96-well plates in 90–200 μL media. All experiments were repeated three times.

### 2.2. Drugs, and Treatments

Phosphoinositide 3-kinases (PI3K) inhibitor BYL719, fibroblast growth factor receptor (FGFR) inhibitor JNJ-42756493, Cyklin-Dependent-Kinase-4/6 (CDK4/6) inhibitor PD-0332991, Poly-ADP-ribose-polymerase (PARP) inhibitors BMN-673 and WEE1 inhibitor MK-1775 were all obtained through Selleckhem Chemicals Munich, Germany. Stocks solution at 10 mM for all inhibitors were initially diluted in the presence of 1% DMSO. Further dilutions were all done with PBS and stored at −20 °C. Before use the inhibitors were diluted once again and used in the following doses: JNJ-42756493 0.01–10 μM; BYL719 0.5–20 μM; PD-0332991 5–40 μM; and for BMN-673 0.1–50 μM and MK-1775 0.1–50 μM.

### 2.3. WST-1 Viability Assay

Viability was analyzed by a WST-1 assay (Roche Diagnostics, Mannheim, Germany) and followed for 72 h (h) after the treatment according to the instructions of the manufacturer and repeated three times and presented as described in more detail before [34].

### 2.4. Cell Proliferation and Cytotoxicity Assays

Cells in 96-well plates were placed into the IncuCyte S3 Live^®^ Cell Analysis System using the Incucyte™ Cytotox Red Reagent (Essen Bioscience, Welwyn Garden City, UK), and followed for proliferation cytotoxicity, taking images every 2 h [34]. Of three repeated experiments a representative experiment was presented.

### 2.5. Statistical Analysis

To verify efficacy of single inhibitors or their combinations compared to the negative control, a multiple *t*-test accompanied by correction for multiple comparison of the means conferring to the Holm Sidak method was done [34,39,40]. The combined effects were evaluated applying the effect-based approach ‘Highest Single Agent’ [39,40], where details have been presented previously [34,39,40].

## 3. Results

### 3.1. Viability after Single Exposure to FDA Approved CDK4/6, PI3K, FGFR, PARP Inhibitors (PD-332991, BYL719, JNJ-42756493, BMN-673 Resp) and a WEE-1 Inhibitor (MK-1775) Measured by WST-1 Assays in HPV^+^ and HPV^−^ TSCC Lines

To test the effects of CDK4/6, FGFR, PI3K, PARP and WEE1 inhibitors, the effects of PD0332991, BYL719, JNJ-42756493, BMN-673 and MK-1775 resp. on the HPV^+^ CU-OP-2, CU-OP-3 and CU-OP-20 and HPV^−^ CUOP-17 cell lines, were examined by WST-1 viability assays. Cell viability was measured after 24, 48 and 72 h with single treatments with the indicated doses below. For BYL719 and JNJ-42756493 only two low doses were included, since higher doses have shown strong inhibition and have been published previously together with the IC50 values of both of the inhibitors [34].

#### 3.1.1. PD-0332991

All CU-OP lines showed dose-dependent responses to 5–40 μM of PD-0332991, with CU-OP-20 being most sensitive and CU-OP-2 most resistant (Figure 1A–D). The highest dose i.e., 40 μM decreased viability completely 24–72 h after treatment in all cell lines (at least *p* < 0.001) and during the same period a >50% decrease in viability was also found in all cell lines except CU-OP-2 with 20 μM PD-0332991 (in all relevant cell lines at least *p* < 0.01). With 10 μM PD-0332991 only CU-OP-20 presented a significant 50% decrease of viability at all time points (at least *p* < 0.05), while CU-OP-3 and CU-OP-17 only had minor decreases in viability after 72 h (at least *p* < 0.05). The 5μM PD-0332991 dose had no effect on any of the four cell lines.

#### 3.1.2. BYL719

HPV^+^ CU-OP-2, CU-OP-3, and CU-OP-20, and HPV^−^ CU-OP-17 have previously been shown to present dose-dependent responses to 0.5–10 μM BYL719 [34]. Here only two very low BYL719 doses were utilized (0.5 μM and 1 μM) showing only marginal decreases in viability (Figure 1E–H). All HPV^+^ cell lines were sensitive to 1 μM BYL719 and had a decrease in viability, except CU-OP-17 (for all indicated, at least *p* < 0.05 after 48 h), while with 0.5 μM BYL719, CU-OP-3 was the only cell line that showed a statistically significant decrease in viability (with at least *p* < 0.001 at 48 h and *p* < 0.05 at 72 h). In contrast, HPV^−^ CU-OP-17 tended here to be resistant to 0.5 μM and 1 μM BYL719.

#### 3.1.3. JNJ-42756493

HPV^+^ CU-OP-2, CU-OP-3, and CU-OP-20, and HPV^−^ CU-OP-17 have previously been shown to present dose-dependent responses to 0.1–10 μM JNJ-42756493 [34]. CU-OP-3, CU-OP-17, and CU-OP-20 were relatively resistant to the two included very low doses (0.1 and 1 μM) of JNJ-42756493 (Figure 1I–L) similar to that reported earlier by us [34]. Only CU-OP-2 (with an FGFR3 mutation) tended to be transiently sensitive with a significant decrease in viability with the highest dose (1 μM) 48 h after treatment (*p* < 0.001).

#### 3.1.4. BMN-673

HPV^+^ CU-OP-2, CU-OP-3, CU-OP-20, and HPV^−^ CU-OP-17 all showed dose-dependent responses to BMN-673 (Figure 2A–D). More specifically, all cell lines were relatively resistant to the three included lower doses (0.1–1 μM) of BMN-673, while the two higher doses 10 and 50 μM were slightly more efficient and gave a decrease viability in all cell lines after 72 h (at least *p* < 0.01). Furthermore, for CU-OP-2 and CU-OP-17, the decrease in viability was statistically significant at all time points when treated with 50 μM BMN-673 (at least *p* < 0.05).

#### 3.1.5. MK1775

HPV^+^ CU-OP-2, CU-OP-3, CU-OP-20, and HPV^−^ CU-OP-17 all showed dose-dependent responses to MK1775 as shown in Figure 2E–H. More specifically, most cell lines were relatively resistant to the lowest included dose (0.1 μM) of MK-1775, with the exception possibly of CU-OP-3, while they all responded better to all the higher doses (1–50 μM at 48 and 72 h after treatment, for all at least *p* < 0.05 (Figure 2E–H). CU-OP-17 was the only cell line that showed a significant response to all treatments 24–72 h (at least *p* < 0.05).

#### 3.1.6. IC50 Values of PD-0332991, BMN-673, and MK-1775 Inhibitors on the CU-OP Cell Lines

For all four CU-OP cell lines the IC50 values for PD-0332991, BMN-673, and MK-1775 are presented in more detail below in Table 1. The IC50 values for BYL719 and JNJ-42756493, for the four CU-OP cell lines, have been published before [34].

More specifically, for PD-0332991 the IC50 values ranged between 9.8 μM and 35.9 μM for the CU-OP cell lines (Table 1). CU-OP-2 consistently had the highest IC50s, while for CU-OP-20 lower concentrations were consistently much more effective, in line with the data shown in Figure 2.

For BMN-673, the IC50 values were between 4.8 μM and extrapolated to 3219 with HPV^−^ CU-OP-17 being the most sensitive and CU-OP 2 and 3 being the most resistant cell lines (Table 1). For MK-1775 the corresponding ranges were 0.2 μM to 9.6 μM, with CU-OP-20 being initially most sensitive, while the others showed relatively similar values.

To summarize these three new inhibitors, CU-OP-20 was generally the most sensitive cell line, with the only exception of BMN-673, where the HPV^−^ CU-OP-17 cell line was more sensitive. The most resistant cell line was generally CU-OP-2, especially when considering the time points 48 and 72 h.

### 3.2. Viability after Combined Exposure of FDA Approved CDK 4/6, PI3K, and FGFR Inhibitors (PD-0332991, BYL719, and JNJ-42756493 Resp.) in HPV^+^ and HPV^−^ TSCC Cell Lines

#### 3.2.1. PD-0332991 and BYL719

Combined treatments with PD-0332991 (5–40 μM) and BYL719 (0.5 and 1 μM) of the four CU-OP cell lines are shown in Figure 3A–H.

The highest dose of PD-0332991 (40 μM) combined with 0.5 or 1 μM of BYL719 resulted in a retained decreased cell viability at all time points for all cell lines analogous to using the highest dose (40 μM) of PD-0332991 alone (for all at least *p* < 0.001) (Figure 1A–H).

Notably, CU-OP-3 showed a statistically significant decrease for all remaining combinations of PD-0332991 with BYL719 0.5 μM, as was also the case for CU-OP-20, except with the BYL719 0.5 μM and the PD-0332991 5 μM combination at 24 and 72 h (for all others at least *p* < 0.05) (Figure 3B,C).

All PD-0332991 doses together with 1 μM BYL719 (Figure 3E–H), significantly reduced cell viability after 72 h for all cell lines, except for CU-OP-2 for the combination of PD-0332991 20 μM with BYL719 0.5 μM (for all others at least *p* < 0.001) (Figure 3E–H).

#### 3.2.2. PD-0332991 and JNJ-42756493

Combined treatments with PD-0332991 (5–40 μM) and 0.1 or 1.0 μM JNJ-42756493 are presented in Figure 3I–P. In all CU-OP cell lines, 40 μM PD-0332991 and 0.1 μM or 1 μM JNJ-42756493 retained a significant decrease in viability analogous to 40 μM PD-0332991 alone (at least *p* < 0.001). Similar observations were made in all cell lines in all with 20 μM PD-0332991 and 0.1 μM or 1 μM JNJ-42756493 combinations, except for CU-OP-2 and CU-OP-17 at 24 h (for all at least *p* < 0.01). However, none to minimal effects on cell viability were observed in all cell lines after 72 h with the lower combinational doses.

#### 3.2.3. Combinational Indexes with the “Highest Single Agent” Approach

The combinational indexes (CIs) of PD-0332991 with BYL719 or JNJ-42756493 were calculated according to the “highest single agent “approach for all cell lines 48 and 72 h after treatment. The CIs after 48 h are shown for specific doses as indicated below for the CU-OP cell lines (Figure 4). The CIs after 72 h generally showed similar trends (data not shown).

*PD-0332991 and BYL719.* Combinations of all doses of the CDK4/6 inhibitor PD-0332991 with the lowest concentration of BYL719 0.5 μM resulted in synergistic, or mainly positive or neutral combinational effects (CI < 1) after 48 h, for CU-OP-3, CU-OP-20, and CU-OP-17, while this was not as much the case for CU-OP-2, that also showed a marginal negative effect (Figure 4A). With the BYL719 1μM dose, all combinations with PD-0332991 mainly resulted in positive or neutral effects (CI < 1 or CI = 1) for all cell lines after 48 h. (Figure 4A).

To summarize, combining the CDK4/6 inhibitor with the PI3K inhibitor, generally resulted in a positive or neutral combinational effect in most cell lines except occasionally for CU-OP-2.

*PD-0332991 and JNJ-42756493.* Combinations with the different doses of the CDK4/6 inhibitor PD-0332991 and the two doses of the FGFR inhibitor JNJ-42756493 are shown in (Figure 4B). Mainly positive or neutral combinational effects (CI < 1) were found when combining PD-0332991 and JNJ-42756493 after 48 h especially for CU-OP-3 and CU-OP-20, while both CU-OP-2 and CU-OP-17 showed negative or neutral effects after 48 h.

To summarize, combining the CDK4/6 inhibitor with the FGFR inhibitor, resulted in mainly neutral, but also some synergistic and negative combinational effects that also tended to be cell line specific.

### 3.3. Viability after Combined Exposure of an FDA Approved PARP and a WEE1 Inhibitor (BMN-673 and MK-1775 Resp), in HPV^+^ and HPV^−^ TSCC Cell Lines

#### 3.3.1. BMN-673 and MK-1775

Combined treatments with some of the lower doses of BMN-673 (0.5 and 1 μM) and MK-1775 (0.1, 0.5, and 1.0 μM) on the four CU-OP cell lines are shown below. A prominent decrease in viability was demonstrated for all four CU-OP cell lines for all BMN-673 and MK-1775 dose combinations at 48 and 72 h after treatment, except for CU-OP-2 after 72 h with the lowest dose combinations (at least *p* < 0.05) (Figure 5A–L).

At 24 h after treatment, most combinations except with the lowest dose of MK-1775 (0.1 μM) also showed a tendency for a decrease in viability, but these did not reach statistical significance (*p* = ns).

To summarize, combining the PARP and WEE1 inhibitors did generally not show superior effects to using one of the drugs alone, with the exception of CU-OP-2, where an increased sensitivity was obtained.

#### 3.3.2. Combinational Indexes with the “Highest Single Agent” Approach

The combinational indexes (CIs) of BMN-673 with MK-1775 were calculated according to the “highest single agent “approach for all cell lines 48 and 72 h after treatment. The CIs after 48 h are shown for specific doses as indicated below for the CU-OP cell lines (Figure 6). The CIs after 72 h generally showed similar trends (data not shown).

For CU-OP-2 all the CIs < 1, indicated positive effects, and this tended to be also the case for the combinations including the 0.1 μM MK-1775 dose for CU-OP-20 and CU-OP-17, but not for CU-OP-3, where the tendency was mainly towards the negative side.

To summarize, for CU-OP-2 positive and synergistic effects were disclosed when combining BMN-673 with MK-1775, while for CU-OP-20 and CU-OP-17 the effects were relatively neutral, while for CU-OP-3 the effects were in the near neutral-negative side.

### 3.4. Proliferation and Cytotoxicity Responses after Single Treatments with FDA Approved CDK4/6, PI3K, FGFR, PARP, and a WEE1 Inhibitors (PD-0332991, BYL719, JNJ-42756493, BMN-673, and MK-1775 Resp.) of HPV^+^ and HPV^−^ TSCC Cell Lines

The efficacy of single treatments with PD-0332991, BYL719, JNJ-42756493, BMN-673, and MK-1775 on the CU-OP cell lines was examined further with regard to proliferation and cytotoxicity, all with PBS as a positive control, and the data are depicted below.

#### 3.4.1. Proliferation

*PD-0332991.* The two lowest doses 5 μM and 10 μM of PD-0332991 used above in the viability tests were shown to inhibit proliferation in all CU-OP cell lines to some extent and in some cases already 24 h after the treatment depending on the cell line (Figure 7A–D). These doses were later used to be tested in combination with the two lowest doses of BYL719 and JNJ-42756493.

*BYL719.* Single treatment with BYL719 at the low dose concentrations (0.5 and 1.0 μM) induced a slight or no decrease in cell proliferation depending on the specific cell line in analogy to the viability data shown earlier (Figure 7E–H).

*JNJ-42756493.* Single treatments with JNJ-42756493 at the low dose concentrations (0.1 and 1 μM) used here did not induce a decrease in cell proliferation in analogy to the viability data (Figure 7I–L).

*BMN-673.* Single treatments with BMN-673 at the depicted two low concentrations (0.5 and 1.0 μM) induced slight or no decreases in cell proliferation depending on the cell line in analogy to the viability data (Figure 7M–P).

*MK-1775.* Single treatments with MMK-1775 at the two depicted low concentrations (0.5 and 1.0 μM) induced dose-dependent decreases in cell proliferation depending on the cell line in analogy to the viability data (Figure 7Q–T).

#### 3.4.2. Cytotoxicity

In addition to the proliferation data above, the cytotoxic effects of the inhibitors with the same doses as used for proliferation were analysed using the Cytotox Red Reagent and the data for all four cell lines are shown in Appendix A.

*PD-0332991.* The CDK4/6 inhibitor PD-0332991 showed no major cytotoxicity with the 5 and 10 μM doses on the HPV^+^ CU-OP-2, CU-OP-3, and CU-OP-20 cell lines, while for HPV^−^ CU-OP-17 cell line some cytotoxicity was observed (Appendix A). However, cytotoxicity could be obtained in all four cell lines when treated with 20 μM or 40 μM of PD-0332991 (data not shown).

*BYL719, JNJ-42756493, MK-1775, and BMN-673.* When distributed alone, at the concentrations used, none of these three inhibitors induced any major cytotoxic effects on any of the cell lines (Appendix A).

### 3.5. Proliferation and Cytotoxicity Responses after Combined Treatments with FDA Approved CDK4/6, PI3K, and FGFR Inhibitors (PD-0332991, BYL719, JNJ-42756493) and of PARP and WEE1 Inhibitors (BMN-673 and MK-1775) of HPV^+^ and HPV^−^ TSCC Cell Lines

The effects of combined treatments with PD-0332991, and BYL719 or JNJ-42756493, or BMN-673 and MK-1775 on the CU-OP cell lines were examined further with regard to proliferation and cytotoxicity, all with PBS as positive control, and the data are depicted below (Figure 8) and data not shown.

#### 3.5.1. Proliferation

*BYL719 and PD-0332991*. Combinational effects on proliferation of BYL719 (0.5 and 1 μM and PD-0332991 (5 and 10 μM) were tested on all CU-OP lines. For CU-OP-2, 3, and -20, proliferation was inhibited for all dose combinations tested throughout the whole observation period of 72 h (Figure 8A–C). For CU-OP-17 inhibition of proliferation was not quite as efficient analogous to the viability data, however, the combination of 10 μM PD-0332991 with 1 μM BYL719 gave the best effect (Figure 8D).

*PD-0332991 and JNJ-42756493.* Combinations of PD-0332991 (5 and 10 μM) with JNJ (0.1 and 1 μM) inhibited proliferation to a higher or lower extent depending on the doses used and the cell line (Figure 8E–H). The most pronounced inhibition of proliferation was observed for the CU-OP-3 and 20 cell lines (Figure 8F,G) in analogy to that observed with the CI data (Figure 4B).

*BMN-673 and MK-1775.* Combinations of BMN-673 (0.5 and 1 μM) and MK-1775 (0.5 and 1 μM) gave a dose-dependent inhibition in all cell lines to some extent (Figure 8I–L) with the most prominent inhibition of CU-OP-2 in analogy to the CI data (Figure 6).

To summarize, upon combining CDK 4/6 inhibitor PD-0332991 with PI3K inhibitor BYL719, an inhibition of proliferation was observed especially for all HPV^+^ CU-OP lines, while the PD-0332991 and FGFR inhibitor JNJ-42756493 combination resulted in inhibition of mainly CU-OP-3 and 20 analogous to the CI data. For the PARP inhibitor, BMN-673 and the WEE-1 inhibitor MK-1775 combination inhibition of proliferation was observed mainly for CU-OP-2 also analogous to the CI data.

#### 3.5.2. Cytotoxicity

*PD-0332991, BYL719, and JNJ-42756493 combinations.* Combining PD-0332991 with BYL719 or JNJ-42756493 at the same doses used in the proliferation tests did not show any marked cytotoxicity for the HPV^+^ CU-OP cell lines (Appendix A). The cytotoxicity observed in the HPV^−^ CU-OP-17 cell line was possibly somewhat enhanced for the PD-0332991 and JNJ-42756493 combination, but not for the PD-0332991 and BYL719 combination, where the cytotoxicity was similar to that with PD-0332991 alone (Appendix A).

*BMN-673 and MK-1775.* Combinations of BMN-673 (0.5 and 1 μM) and MK-1775 (0.5 and 1 μM) did not result in any major cytotoxicity effects although some minor effects were observed for CU-OP-2 and 20 (Appendix A)

## 4. Discussion

Here, the CDK4/6 inhibitor PD-332991, the PI3K inhibitor BYL719, and the FGFR inhibitor JNJ-42756493 were tested alone and combined on HPV^+^ CU-OP-2, CU-OP-3, CU-OP-20, and HPV^−^ CU-OP-17 TSCC cell lines. In addition, PARP and WEE1 inhibitors, BMN-673 and MK-1775 were tested alone or combined on the same cell lines.

All TSCC cell lines showed dose-dependent decreases in viability and proliferation upon single treatments with PD-332991, BYL719, JNJ-42756493, BMN-673, and MK-1775 as shown here or previously [34], but the sensitivity of the cell lines varied depending on the inhibitor and the doses used. None of the above inhibitors showed any marked cytotoxic effects upon single or combined use with the lower doses included here against any of the HPV^+^ CU-OP-2, CU-OP-3, and CU-OP-20 cell lines. However, some cytotoxicity could be observed against HPV^−^ CU-OP-17 with the PD-332991 and BYL719 combination.

Upon combining the CDK4/6 inhibitor PD-332991 with the PI3K inhibitor BYL719, mainly positive or synergistic effects with enhanced inhibition of viability and proliferation were observed in all cell lines, especially for HPV^+^ CU-OP-3, CU-OP-20, and HPV^−^ CU-OP-17. The CDK4/6 inhibitor PD-332991 and JNJ-42756493 combination also gave decreased viability and proliferation for all cell lines, but the effects were more variable. With the latter combination, synergistic effects were less common, with the exception of in CU-OP-3, while for CU-OP-2 and CU-OP-17 mainly antagonistic effects were observed. Finally, when the PARP inhibitor BMN-673 was combined with the WEE1 inhibitor MK-1775, although decreases in viability and proliferation were obtained, the only synergistic effect observed was in the CU-OP-2 cell line.

The fact that dose-dependent responses to the inhibitors were noted when using high doses of the drugs was not unexpected. Commenting first with regard to PD-332991, the effect of the lower dose (5 μM) of the inhibitor on the CU-OP cell lines would characterize these cells as rather resistant to this drug if one compares it with another report [41]. In that study, the HPV^−^ Fadu head neck cancer cell line was shown as relatively sensitive, while two other HPV^+^ head neck cancer cell lines UM-SCC-47 and UP-SCC-154 were similar to the CU-OP cell lines here (and also in an earlier study by us) less sensitive and defined as more resistant [38,41].

Notably, however, by combining 5 μM of PD-332991 with low doses of 0.5 and 1 μM of BYL719, gave for CU-OP-3 a remarkable decrease in the inhibition of viability and proliferation and could in fact be clinically useful, as for example as already initiated and shown for breast cancer [35]. However, despite the promises of using these drugs in combination, the safety of using these drugs alone or combined or in combination with endocrine therapies must also be accounted for and scrutinized on a molecular basis [42,43,44].

Some positive effects were also disclosed when combining PD-332991 with JNJ-42756493. Moreover, for both these latter two combinations, the best effects were observed for CU-OP-3, and although synergistic effects were also observed for all cell lines with the PD-332991 and BYL719 combinations, this was less so with the PD-332991 and JNJ-4256493 mix. For the latter on the other hand, there is to our knowledge not much information in the scientific literature.

With regard to the cytotoxic effects of the above drugs (PD-332991, BYL719, JNJ-4256493, BMN-673, or MK-1775), it was obvious that low doses of all the drugs used here had no major effects on any of the HPV^+^ CU-OP-2, CU-OP-3 or CU-OP-20 cell lines. However, low doses of PD-332991 had a cytotoxic effect on the HPV^−^ CU-OP-17 cell line. Nevertheless, when combining the PD-332991 with BYL719 or JNJ-42756493, or BMN-673 and MK-1775 with the doses used here, a major increase in cytotoxicity was not observed for this cell line or the other ones.

Of note, in this study only, very low doses of BYL719 and JNJ-42756493 were included, since our previous data showed that most of our cells were relatively sensitive to the effects of BYL719 and almost always so also for JNJ-42756493 [34]. Furthermore, when using high doses of these drugs, or corresponding other PI3K or FGFR inhibitors, their efficacy was so good when used alone, that the synergistic effects of combining them could be masked [34,45]. In this study, however, the BYL719 batch that was used was marginally less effective compared to that in one of our previous report [34].

Slightly different from the above inhibitors the sensitivity to BMN-673 showed another pattern, with CU-OP-2 initially being more sensitive than CU-OP-3 in line with a previous study by us where the PARP1 inhibitor AZD2281 reduced colony formation more readily in CU-OP-2 than in CU-OP-3 [38]. Furthermore, in that report, the response to AZD2281 was not correlated to the HPV status of the cell lines, or the initial PARP1 expression of the cell lines, nor was it correlated to the ability of the cell to conduct double-stranded DNA repair (DSB) [38].

For MK-1775, relatively low doses of the drug could be used against all cell lines. This was in accordance with one previous report [46]. There the effects of AZD-1775 (the Astra Zeneca equivalent to the Merk MK-1775) on the HPV^+^ cell line UM-SCC-47 were investigated also in combination with cisplatin and it was shown that even low doses of AZD-1775 could sensitize this cell line to cisplatin through apoptosis [43]. In a more recent study, AZD-1775 was combined with different Chk1 inhibitors and the effects on radio sensitization were examined [44]. In that study, synergistic effects were observed when combining Chk1 inhibitors with the WEE1 inhibitor AZD-1775, and despite using lower doses of the two drugs prominent effects on radio sensitization were disclosed [47].

When we then continued with our investigation we could show that by combining the PARP inhibitor BMN-673 and the WEE1 inhibitor MK-1775 synergy was mainly disclosed for CU-OP-2 and not for CU-OP-3 as presented above for the other combinations. In addition, some positive effects were obtained for CU-OP-20 and CU-OP-17. Here, it is not possible to compare our data to corresponding data from others in head and neck cancer. So far namely, to our knowledge using PARP and WEE1 inhibitors in combination for oropharyngeal or head and neck cancer has still not been pursued, despite positive effects for other cancers, as shown and reviewed by others [36,37,48,49,50].

Analyzing possible mechanisms for the above, obtaining an efficient decrease in cell viability and proliferation in the CU-OP cell lines after treatment with the CDK4/6 inhibitor PD-33291 and combining it with PI3K and FGFR inhibitors, BYL719 and JNJ-42756493 can have several explanations. A strong response after using these combinations could be that both CDK4/6 and FGFR inhibitors affect the PI3K pathway, which could naturally enhance the effect of the inhibitors [51]. Furthermore, the CDK4/6 protein acts in the G1-S point of the cell cycle so the inhibitor might act at this cell cycle checkpoint [48]. More specifically, Cyclin D activates CDK4/6 which phosphorylates the retinoblastoma tumor suppressor protein (pRB) and thereby inhibits its effect on E2F factors. Consequently, pRB phosphorylation allows for activation of E2F factors allowing DNA replication and the entrance of the cell again into the cell cycle [52]. CDK4/6 inhibitors have therefore been suggested to be useful for therapeutic strategy, and by inhibiting CDK4/6, pRB could function and block E2F and interrupt cell cycle progression [53]. In breast cancer, the CDK4/6 inhibitor Palbociclib, has already been successfully used for the treatment of oestrogen receptor-positive metastatic breast cancer and other breast cancer subtypes [54,55]. Furthermore, the potential positive clinical role of Palbociclib is suggested in combination with EGFR inhibitor Cetuximab in metastatic and advanced HPV unrelated head and neck cancer [56]. The latter has also been supported experimentally in HPV^−^ cell lines and also discussed in a relatively recent review [57,58].

The fact that there were synergistic effects using the PARP inhibitor BMN-673 together with the WEE1 inhibitor MK-1775 was not completely unexpected. PARP can namely stabilize reversed forks during DNA replication and prevents fork collapse and the inhibition of PARP enhances stress on the replication forks and may be useful also on cells with high replicative stress in general [36]. Cells that then undergo replicative stress depend on WEE1 kinase, a key regulator of the G2 to M checkpoint. By inhibiting also WEE1 G2 to M is abrogated directing the cells with a deregulated G1 into premature mitosis resulting in mitotic catastrophe or apoptosis and cell death [36,37].

In this study, we found that the CU-OP-3 cell line was more sensitive to the synergistic effects of the CDK4/6 and PI3K and FGFR inhibitors, while the CU-OP-2 cell line was more sensitive to the latter PARP and WEE1 inhibitor combination. Why this is the case we do not presently know. The CU-OP-2 line harbors both a *PIK3CA* and *FGFR3* mutation and should therefore theoretically be more sensitive to the PI3K and FGFR inhibitors, as actually is the case when compared to CU-OP-3 with the low doses used in this study. It is also notable that CU-OP-20 with a *PIK3CA* mutation but not an *FGFR3* mutation is also somewhat sensitive to the PI3K inhibitor, but not to the FGFR inhibitor at the low doses used herein. We, therefore, suggest that is possible that a higher sensitivity to a single dose of an inhibitor may mask the synergy obtained as sometimes noted previously [42]. However, the fact that by combining different inhibitors one can obtain synergy against relatively resistant cell lines such as CU-OP-2 and CU-OP-3, that also are both radioresistant [59] is encouraging for the pursuit of personalized medicine. Nonetheless, more information on potential confounding by different genetic backgrounds, or additional mutations, would be of value to pursue in future studies.

There are limitations to this study. Despite the many cell lines used, one could suggest that even more cell lines should be tested. Furthermore, presently we have not conducted in vivo experiments on all cell lines tested so far, nor have we gone into detail with regard to molecular mechanisms or signaling pathways affected in the different cell lines. For the future one would be interested in being able to upfront have the possibility to predict which tumors would be eligible for a specific inhibitor combination. However, we are presently not there yet, but obviously, this will be one of our future goals.

To conclude, combinations of CDK 4/6 with PI3K or FGFR inhibitors, especially the former two disclose mainly positive and synergistic effects, but also some negative effects on the decrease of cell viability and the inhibition of cell proliferation of HPV^+/−^ TSCC cell lines. In addition, in some cases, PARP1 and WEE1 inhibitors combined can present synergistic effects on specific HPV^+^ TSCC cell lines. Combining these different inhibitors could be a beneficial therapeutic opportunity in specific cases and warrants further investigations to better stratify which approaches are best for specific tumors with specific molecular profiles.

## 5. Conclusions

All tested CDK4/6, PI3K, FGFR, PARP, and WEE1 inhibitors presented dose-dependent inhibitory effects on all tested TSCC lines. Synergy was mainly obtained when combining CDK4/6 with PI3K inhibitors, but less frequently obtained when combining CDK4/6 with FGFR inhibitors or PARP with WEE1 inhibitors and varied also depending on the individual cell lines.

## Figures and Tables

**Figure 1 viruses-14-01372-f001:**
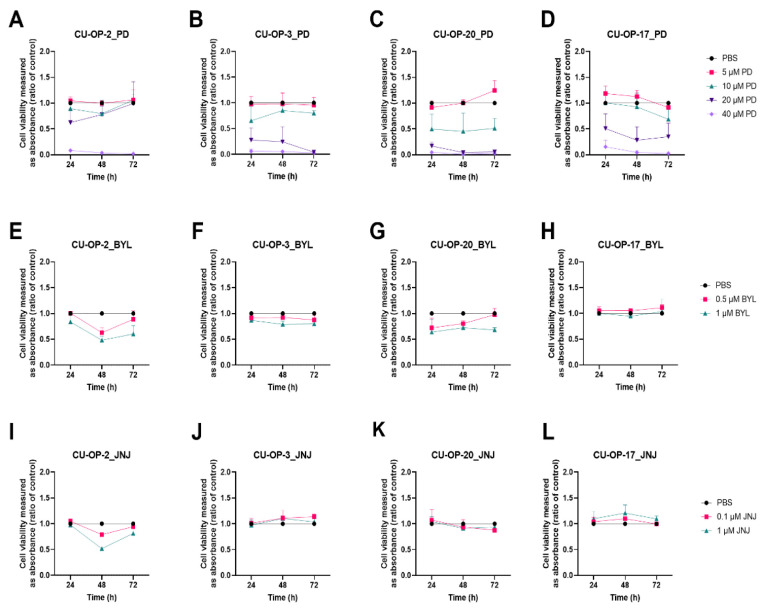
WST-1 viability assay on HPV^+^ cell lines CU-OP-2, CU-OP-3 and CU-OP-20 and HPV^−^ cell line CU-OP-17. Cell viability analysis measured as absorbance after 24, 48 and 72 h of single treatment with PD-0332991 (**A**–**D**), BYL719 (**E**–**H**), JNJ-42756493 (**I**–**L**). PD denotes PD-0332991; BYL denotes BYL719; and JNJ denotes JNJ-42756493.

**Figure 2 viruses-14-01372-f002:**
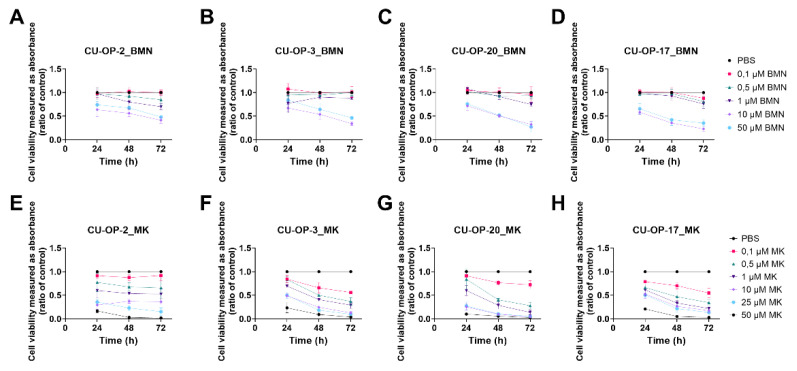
WST-1 viability assay on HPV^+^ cell lines CU-OP-2, CU-OP-3, and CU-OP-20, and HPV^−^ cell line CU-OP-17. Cell viability analysis measured as absorbance after 24, 48, and 72 h of single treatments with BMN-673 (**A**–**D**), or MK-1775 (**E**–**H**). BMN denotes BMN-673 and MK denotes MK-1775.

**Figure 3 viruses-14-01372-f003:**
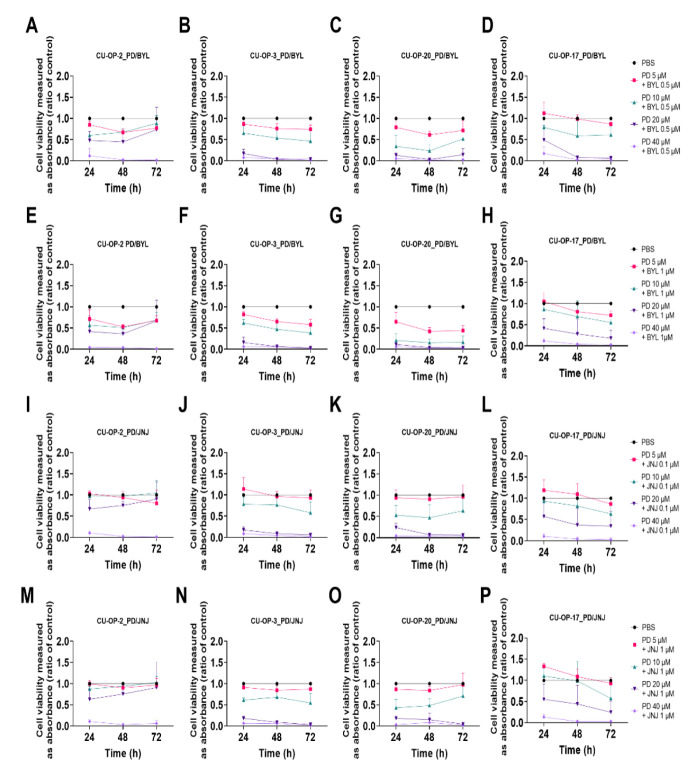
WST-1 viability assay on HPV^+^ cell lines CU-OP-2, CU-OP-3, and CU-OP-20, and HPV^−^ cell line CU-OP-17. Cell viability analysis measured as absorbance after 24, 48, and 72 h of combination treatment with PD-0332991 and BYL719 (**A**–**H**) or PD-0332991 and JNJ-42756493 (**I**–**P**). PD denotes PD-0332991; BYL denotes BYL719; and JNJ denotes JNJ-42756493.

**Figure 4 viruses-14-01372-f004:**
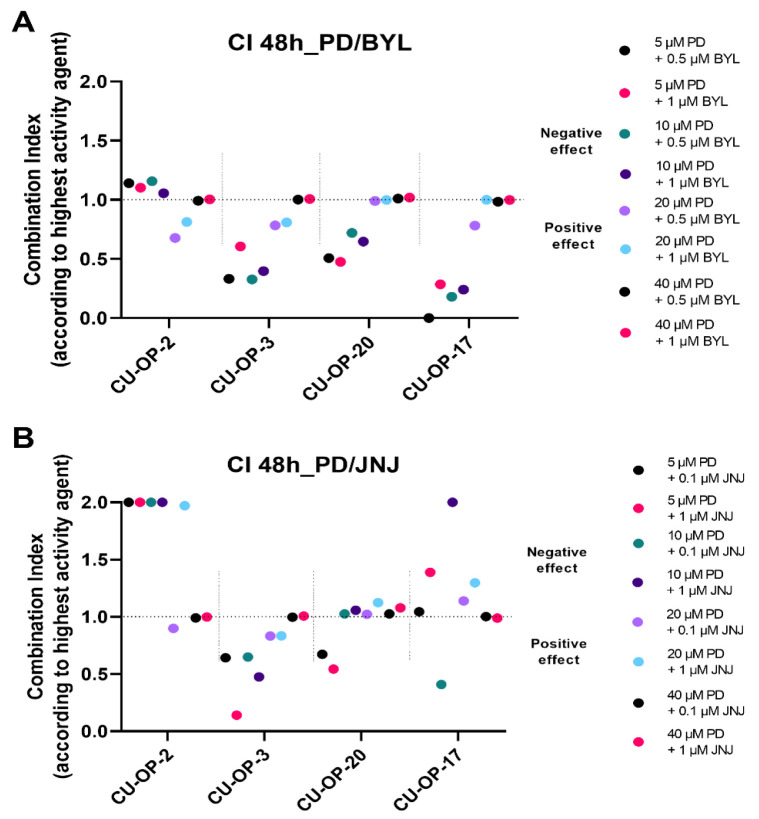
Combinational effects of CDK4/6 inhibitor PD-0332991 and PI3K inhibitor BYL719 (**A**) and PD-0332991 and FGFR inhibitor JNJ-42756493 (**B**) after 48 h. Combination indexes (CIs) were shown with the highest single agent approach after 48 h (**A**,**B**) and 72 h, where CI > 1 shows a negative combination effect and CI < 1 shows a positive combination effect. CIs were calculated from the mean of three experiments analyzed by WST-1, at 48 h after treatment. CI denotes combinational index; PD denotes PD-0332991; BYL denotes BYL719; and JNJ denotes JNJ-42756493.

**Figure 5 viruses-14-01372-f005:**
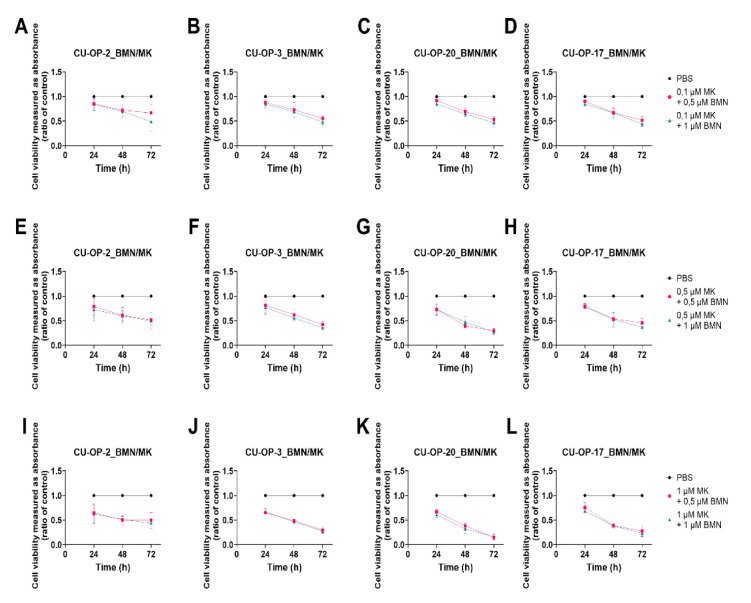
WST-1 viability assay on HPV^+^ cell lines CU-OP-2, CU-OP-3, and CU-OP-20 and HPV^−^ cell line CU-OP-17. Cell viability analysis measured as absorbance after 24, 48, and 72 h of combination treatment with BMN-673 and MK-1775 (**A**–**L**). BMN denotes BMN-673 and MK denotes MK-1775.

**Figure 6 viruses-14-01372-f006:**
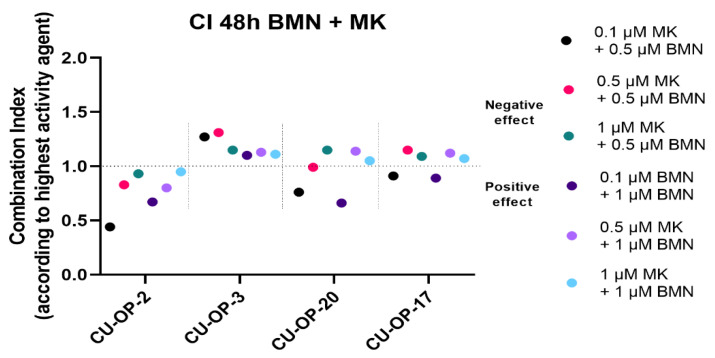
Combinational effects of PARP inhibitor BMN-673 and WEE1 inhibitor MK-1775 after 48 h. Combination indexes (CIs) were shown with the highest single agent approach after 48 h, where CI > 1 shows a negative combination effect and CI < 1 shows a positive combination effect. CIs were calculated from the mean of three experiments analyzed by WST-1, at 48 h after treatment. CI denotes combinational index; BMN denotes BMN-673 and MK denotes MK-1775.

**Figure 7 viruses-14-01372-f007:**
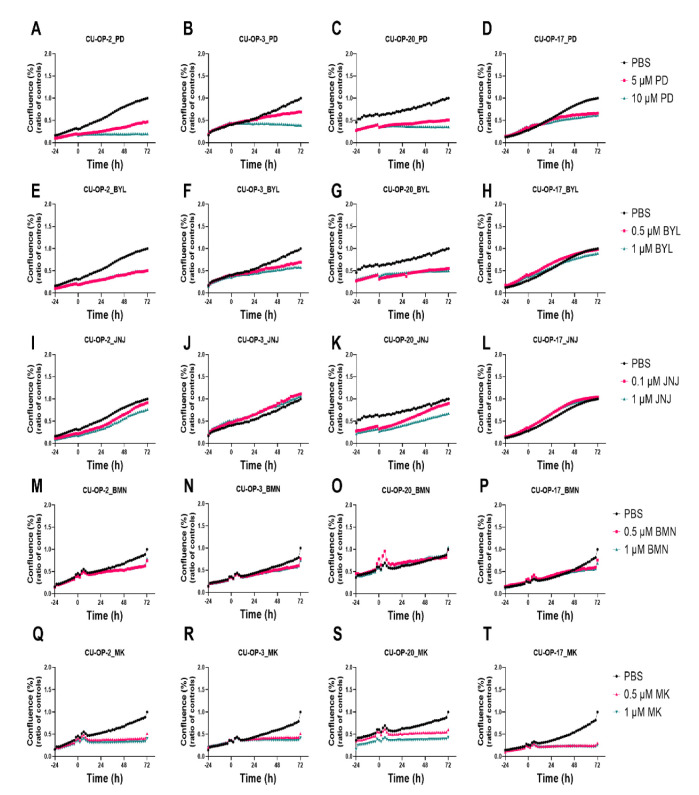
Proliferation response of HPV^+^ CU-OP-2, CU-OP-3, CU-OP-20, and HPV^−^ CU-OP-17 cell lines upon treatment with PD-0332991, BYL719 or JNJ-42756493, BMN-673, and MK-1775. Proliferation responses of HPV^+^ CU-OP-2, CU-OP-3, CU-OP-20, and HPV^−^ CU-OP-17 after treatment with CDK4/6 inhibitor PD-0332991 (**A**–**D**), PI3K inhibitor BYL719 (**E**–**H**), FGFR inhibitor JNJ-42756493 (**I**–**L**), PARP inhibitor BMN-673 (**M**–**P**) and WEE1 inhibitor MK-1775 (**Q**–**T**). The graphs represent one experimental run per cell line. Confluence (%) denotes proliferation response; BYL denotes BYL719; JNJ denotes JNJ-4275649 and PD denotes PD-0332991; BMN denotes BMN-673; MK denotes MK-1775.

**Figure 8 viruses-14-01372-f008:**
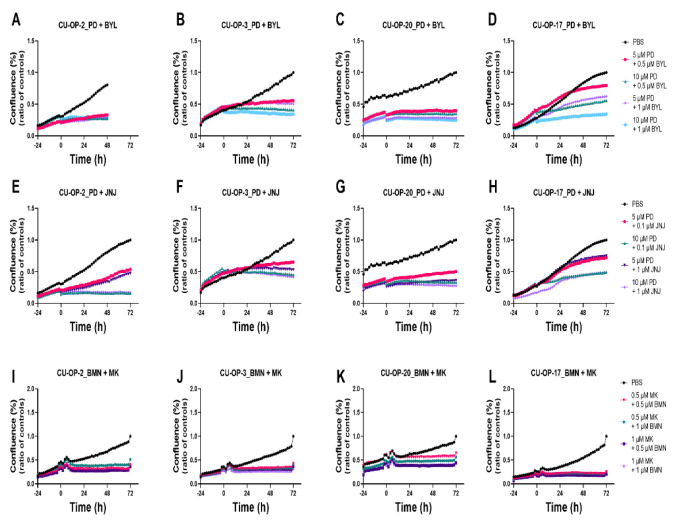
Proliferation response of HPV^+^ CU-OP-2, CU-OP-3, CU-OP-20, and HPV^−^ CU-OP-17 cell lines upon combined treatment with PD-0332991 and BYL719 or JNJ-42756493. Proliferation responses of HPV^+^ CU-OP-2, CU-OP-3, CU-OP-20, and HPV^−^ CU-OP-17 after treatment with CDK4/6 inhibitor PD-0332991 and PI3K inhibitor BYL719 (**A**–**D**) or with FGFR inhibitor JNJ-42756493 (**E**–**H**), or with the PARP inhibitor BMN-673 and WEE1 inhibitor MK-1775 (**I**–**L**). The graphs represent one experimental run per cell line. Confluence (%) denotes proliferation response; PD denotes PD-033299; BYL denotes BYL719; JNJ denotes JNJ-4275649; BMN denotes BMN-673; and MK denotes MK-1775.

**Table 1 viruses-14-01372-t001:** Estimation of inhibitory concentration 50% (IC50) based on WST-1 viability assay following treatment with the CDK4/6 inhibitor (PD-0332991), with the PARP inhibitor (BMN-673) and with the WEE1 inhibitor MK-1775 at 24, 48 and 72 h after treatment.

IC50 (µM) ^b^				
Drugs	Cell Lines	24 h	48 h	72 h
CDK4/6 (PD-0332991)	CU-OP-2	25.1	26.8	35.9
	CU-OP-3	13.4	15.2	12.3
	CU-OP-20	10.4	9.8	10.1
	CU-OP-17	21.0	16.6	14.4
PARP (BMN-673)	CU-OP-2	291.4 ^a^	127.2 ^a^	14.9
	CU-OP-3	3219 ^a^	81.4 ^a^	13.9
	CU-OP-20	229.5 ^a^	30.9	6.0
	CU-OP-17	93.8 ^a^	12.8	4.8
WEE1 (MK-1775)	CU-OP-2	3.4	2.4	1.7
	CU-OP-3	9.6	0.6	0.2
	CU-OP-20	2.9	0.4	0.2
	CU-OP-17	6.5	0.4	0.2

The inhibitory concentration 50% (IC50) for each cell line for each drug was determined from log concentrations effect curves in GraphPad Prism using non-linear regression analysis. ^a^ Extrapolated IC50 value, i.e., outside the tested concentration range. ^b^ IC50 values for BYL and JNJ previously calculated and presented (34).

## Data Availability

Not applicable.

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
