# Peer review of "Targeted Therapy of HPV Positive and Negative Tonsillar Squamous Cell Carcinoma Cell Lines Reveals Synergy between CDK4/6, PI3K and Sometimes FGFR Inhibitors, but Rarely between PARP and WEE1 Inhibitors"

_viruses, 2022, doi:10.3390/v14071372_

Round 1
Reviewer 1 Report
In this manuscript, the authors evaluated the effect PI3K, CDK4/6 or the FGFR inhibitors combined or alone with PARP and WEE1 inhibitors in several TSCC cell lines. This manuscript extends the previous results published in Holzhauser et al., Front Oncol 2021 and showed a beneficial therapeutic opportunity of inhibitors combination in specific cell lines tested.
I have only a few comments:
How it is with PIK3CA and FGFR3 mutation in HPV negative cell lines? It is not very clear in section Materials and methods.
Figures are very small and not easily readable.
Author Response
I have only a few comments:
How it is with PIK3CA and FGFR3 mutation in HPV negative cell lines? It is not very clear in section Materials and methods.
According to the information we have, this cell line does not have any PIK3CA or an FGFR3 mutation, neither does CU-OP-3. The brackets after each cell line (CU-OP-2 and CU-OP-20) denotes only the cell lines that have such mutations
Figures are very small and not easily readable.
We thank the reviewer for this comment and have tried to increase the size of several figures, especially Figures, 3, 5, 7 and 8.
Reviewer 2 Report
HPV+ head and neck cancers are increasing in many countries. While these cancers frequently show more favorable response to therapy and different mutational and transcriptomic profiles they are currently treated similarly to patients with HPV- HNSCC. The authors here test FDA approved treatments against CDK4/6, PARP, and WEE1 for their individual and combinatorial efficacy in restricting the viability and/or proliferation/cytotoxicity of three HPV+ HNSCC cell lines and one HPV- HNSCC (they appear to have done more work using these same cell lines with different treatments, which is briefly discussed in the current manuscript). The results are presented in a consistent manner and their broader conclusion that in most cases there appears to be a benefit in at least some of the treatment combinations is certainly interesting and may contribute toward shifting the treatment paradigm. Nevertheless, the beneficial treatment combinations do not appear to be always predicted by the previously described mutational profile of the cell lines and thus preclude more broadly unifying conclusions. While the authors mention this topic in their discussion they should further address the potential confounding by different genetic backgrounds as well as discuss potential ways forward in light of these results.
Author Response
We thank the reviewer for the review and more specifically the last sentence of the reviewers comment.
We have added a sentence in the discussion on page 14, lines 541-542 to address this important issue. This topic, needs definitely more research and we hope to embark on such studies in the near future.
Reviewer 3 Report
Tonsillar and base of tongue squamous cell carcinoma (TSCC/BOTSCC) are often caused by HPV. The incidence of these cancers has increased in many Western countries. HPV-positive cancers have generally better prognosis than the corresponding HPV-negative cancers, but upon relapse, survival is poor.
Kostopoulou et al. have explored the use of CDK4/6, PI3K, PARP and WEE1 inhibitors in different combinations on tonsillar and base of tongue squamous cell carcinoma (TSCC) cell lines.
Viability, proliferation, and cytotoxicity were followed by WST-1 assays and the IncuCyte S3 Live® Cell Analysis System. All inhibitors presented dose dependent inhibitory effects on tested TSCC lines. Synergy was mainly obtained when combining CDK4/6 with PI3K inhibitors, but less frequently obtained when combining CDK4/6 with FGFR inhibitors or PARP with WEE1 inhibitors and varied also depending on the individual cell lines.
The claims are properly placed in the context of the previous literature. The experimental data support the claims. The manuscript is written clearly enough that most of it is understandable to non-specialists. The authors have provided adequate proof for their claims, without overselling them. The authors have treated the previous literature fairly. The paper offers enough details of methodology so that the experiments could be reproduced. The paper is well written and interesting to read.
Author Response
We thank the reviewer for this review.